# Exponentially Expanding the Compiler Phase-Ordering Problem's Search Space through the Learning of Dormant Information

## Abstract

Applying compilation transformations in optimal sequences can significantly improve program speed and reduce code size. However, finding these optimal sequences, a problem known as the phase-ordering problem, remains a longstanding challenge. Specifically, modern compilers offer hundreds of available transformations, making the search space too large to efficiently explore in a reasonable time frame. Existing solutions tackle this problem by grouping transformations into short sequences, based on prior knowledge from human experts, and then searching for optimal orders within these sequences. However, such pruning methods can be **aggressive**, potentially excluding optimal solutions from the search space. Additionally, they lack scalability for new transformations.

In this paper, we propose a new, more **conservative** pruning approach that relies on Machine Learning to capture dormant information. This approach only excludes non-optimal solutions from the search space. It does not rely on any prior human knowledge, making it scalable to new transformations.

To demonstrate the efficiency of this conservative pruning approach, we integrate it with a classical Reinforcement Learning model, previously used with aggressive pruning methods. Our solution, named FlexPO, is capable of exploring a search space that is exponentially larger than those considered in existing solutions. Experimental results demonstrate that FlexPO generates programs that are 12% faster or 17.6% smaller than the programs generated by modern compilers.

## 1 Introduction

Modern compilers offer hundreds of compilation transformations to modify programs. Extensive research Ashouri et al. (2016a); Kulkarni & Cavazos (2012); Ashouri et al. (2017); Kulkarni et al. (2003) has demonstrated that applying these transformations in an optimal order can lead to significant improvements in program runtime, ranging from 2.4% to 60.37%.

However, the diverse effects of different transformations and the intricate interactions among them pose challenges in determining the optimal sequences for specific programs. To mitigate this, modern compilers offer predefined transformation sequences, such as -O3, -Oz, and -Os, which are based on empirical studies. Despite this, research suggests that these predefined sequences have room for improvement Gong et al. (2018); Mammadli et al. (2020); Ashouri et al. (2017); Jain et al. (2022). Applying a one-size-fits-all sequence to different programs can result in suboptimal outcomes, indicating that these sequences may be too generic to achieve optimal performance.

The phase-ordering problem, which involves selecting and applying transformations in optimal sequences, remains an unsolved challenge. This complexity arises from the enormous and intricate search space, which expands exponentially with the number of transformations. For $N$ available transformations, the search space contains $N^L$ potential sequences when applying $L$ transformations. Predicting the outcomes of these sequences or assessing the similarity between the outputs of different sequences is particularly challenging, due to the complex interactions among transformations Gong et al. (2018); Whitfield & Soffa (1997).

Some researchers Ashouri et al. (2017); Jain et al. (2022); Mammadli et al. (2020) propose a pruning mechanism to reduce the search space. This approach first leverages prior human knowledge to

cluster transformations into short sequences. It then employs search algorithms to select and order these sequences. Compared to ordering individual transformations, ordering sequences significantly reduces the search space, allowing the search process to find good solutions in a reasonable time frame. However, this approach has two limitations. First, the pruning mechanism is **aggressive**, potentially excluding optimal solutions from the pruned search space. Second, the mechanism relies on human experts to form the transformation sequences, making it **less scalable** for incorporating new transformations. In this paper, we refer to this mechanism as *aggressive pruning*.

In this paper, we propose a new pruning mechanism. This mechanism uses an ML model to learn how to detect transformations that will not change the program (a.k.a. dormant transformations). It then guides the search process to focus solely on transformations that are likely to change the program (a.k.a. active transformations). Compared to the aggressive pruning solution, the new mechanism is **conservative**, as it only prunes non-optimal sequences from the search space. Furthermore, it offers **scalability** for new transformations, as it does not rely on prior knowledge of the transformations.

The proposed conservative pruning can replace the existing aggressive pruning mechanism. Specifically, we introduce FlexPO, a framework that integrates conservative pruning with a classical RL model. This RL model has been used to solve the phase-ordering problem in conjunction with aggressive pruning mechanisms Jain et al. (2022); Ashouri et al. (2017); Mammadli et al. (2020). With the new pruning approach, FlexPO can explore a search space that is exponentially larger than those of other solutions, as shown in Figure 1.

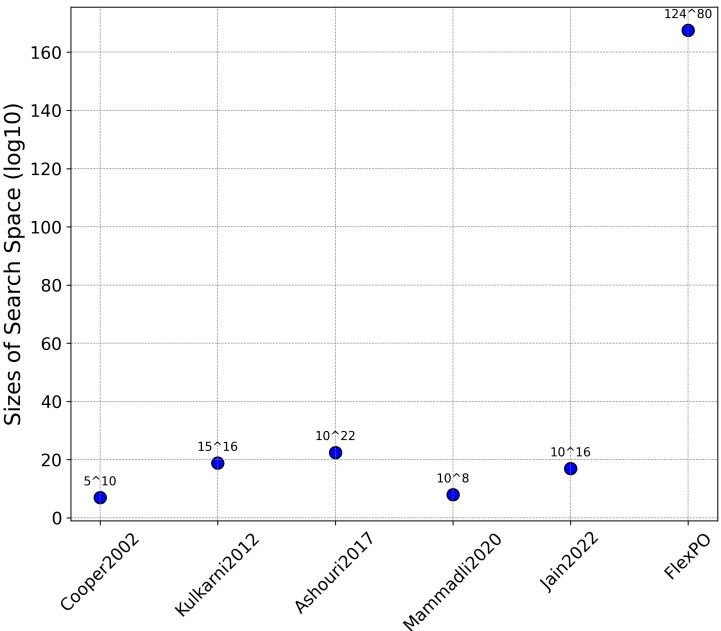

Figure 1: FlexPO supports a search space that is exponentially larger than existing solutions.

The contributions of our paper are listed below:

- Introduce a new mechanism for representing programs using dormant information;
- Propose the use of an ML model to learn the dormant status of transformations and apply conservative pruning during the phase-ordering problem;
- Replace the aggressive pruning mechanisms in existing solutions with conservative pruning, and evaluate the improvements using popular C++ applications.

## 2 RELATED WORK

The phase-ordering problem has long posed a significant challenge for researchers due to the extensive and intricate search space it involves. Early attempts to address this problem using predictive heuristics

have proven to be suboptimal, particularly as the number of transformations increased and their interactions became more complex Triantafyllis et al. (2003). In recent years, the application of Machine Learning (ML) models has garnered attention for solving this problem. Researchers have explored various approaches, including Genetic Algorithms Cooper et al. (1999; 2002); Kulkarni et al. (2004); Martins et al. (2016), Markov models Agakov et al. (2006), and Recommendation Systems Ashouri et al. (2017), to search for optimal transformation sequences. Reinforcement Learning (RL) has also emerged as a promising avenue for research. Mammadli et al. Mammadli et al. (2020) tried to utilize RL model to address the phase-ordering problem. However, their framework faced difficulties in identifying good sequences within a reasonable time frame due to the expansive search space.

The current state-of-the-art solutions for the phase-ordering problem often employ aggressive pruning mechanisms. These approaches involve clustering transformations into short sequences that have shown good performance, and then searching for the optimal arrangement among these sequences. This strategy effectively reduces the search space. For example, in the original formulation of the phase-ordering problem where there are $N$ transformations and a desired sequence length of $L$, the search space consists of $N^L$ possible sequences. However, by clustering $G$ different transformations into sequences, we are left with $\frac{N}{G}$ such sequences. Consequently, only $\frac{L}{G}$ of these sequences need to be selected to form the final solution. This reduces the search space to $\left(\frac{N}{G}\right)^{\frac{L}{G}}$.

There are two main challenges associated with aggressive pruning solutions. First, these methods typically rely on prior knowledge to cluster transformations into short sequences. Projects such as POSET-RL Jain et al. (2022) and MICOMP Ashouri et al. (2017) use existing $O3$ or $Oz$ pipelines as the basis for creating these sequences. However, this approach has limitations when dealing with transformations not included in the $O3$ or $Oz$ pipelines. Second, evaluations conducted in earlier studies Cooper et al. (2002) have shown that optimal solutions for the phase-ordering problem are sparsely distributed throughout the search space. As a result, aggressive pruning runs the risk of **excluding** these optimal solutions from the pruned search space.

## 3 INSIGHT

Aggressive pruning methods have the risk to exclude optimal solutions during the pruning process. In contrast, we propose a conservative pruning strategy in this paper that aims to retain these optimal solutions within the search space. The fundamental insight behind conservative pruning lies in the observation that certain transformations are dormant for a given program, meaning they do not change the program Kulkarni et al. (2004). For instance, if a program has already undergone the $inline$ transformation, which inlines function calls, applying $inline$ again would likely yield no change. Likewise, if the $loop - unroll$ transformation has been executed and no loop structures remain in the program, additional loop-related transformations may become dormant. Based on this understanding, our conservative pruning approach aims to guide the search process by excluding these dormant transformations. In doing so, we can substantially reduce the search space while ensuring that optimal solutions are not pruned away.

We are **NOT** the first to utilize dormant information for the phase-ordering problem. Work by Kulkarni et al. Kulkarni et al. (2004) analyzes dormant transformations within an evolutionary algorithm to eliminate redundant searches of sequences with identical effects. However, existing solutions do **NOT** treat dormant information as **learnable**. On the contrary, they rely on human expertise to establish rules which are used to identify dormant transformations. As a result, their solutions are not scalable and support only 15 transformations.

We are the first to consider dormant information as learnable, employing an ML model to predict this information and apply it during the phase-ordering search process. Unlike existing solutions, our approach does not depend on human expertise and is scalable to new compilation transformations. We evaluate our method using 124 transformations in Section 7.

## 4 PROGRAM REPRESENTATION BY DORMANT INFORMATION

One of the most significant challenges in applying ML algorithms to program optimization is the representation of programs themselves. Unlike natural language, which follows a linear order,

programs contain a wealth of information that is not easily representable. Some researchers Ben-Nun et al. (2018); Cummins et al. (2020) use graph models to capture program characteristics, particularly control-flow information. However, these methods often involve substantial computational overhead for program analysis. In the context of the phase-ordering problem, it has been observed that transformations are sensitive to specific program features. Consequently, researchers Haj-Ali et al. (2020); Ashouri et al. (2017; 2016b) propose using feature vectors to represent programs. These feature vectors capture specific attributes, such as the number of loops or the number of branch instructions, to better inform the optimization process.

In this paper, we propose a new mechanism to represent programs, which represent programs with activate/dormant status of applied transformation, as shown in Figure 2. The proposed representation maps programs to vectors, the length is same as the number of available transformations. Each element in the vector represents whether the corresponding transformation has been applied, and whether it is dormant if it has been applied. This new representation is used to predict the dormant status of new transformations.

Figure 2: Different programs show different activate/dormant status when applying the same transformations. Thus, we propose to represent programs with the activate history of applied transformation.

The utility of the new representation is based on two observations. First, the dormant status of a transformation is related to specific features of the program. For instance, if all applied loop-related transformations are dormant for a given program, it is likely that the program has few or no loop structures. Second, the dormant statuses of different transformations are correlated with each other. For example, if a constant folding transformation has been activated, it is likely that a subsequent dead code elimination transformation will also be activated. This is because the constant folding transformation may generate dead code that the dead code elimination transformation can remove.

Our new mechanism offers several advantages. First, it is lightweight and easy to capture during the search process. Second, it is scalable and can easily accommodate new transformations proposed in the future. Third, it requires only a scalar vector to store the program representations, thus consuming minimal memory. Fourth, it goes beyond capturing simple program features (e.g., number of loops, number of return instructions) to reflect more complex interactions present within programs. For example, the activation of the loop-unroll transformation suggests that the program contained a loop structure, and the transformed programs are expected to feature numerous similar code sections.

## 5 LEARNING DORMANT INFORMATION

The proposed program representation is used to predict the dormant status of subsequent transformations, enabling the application of conservative pruning. Figure 3 illustrates an example. The input, the proposed program representation, is a vector whose length is equal to the number of transformations. Each element in the vector can be 1, 0, or −1, indicating whether the corresponding transformation has been applied and, if so, whether it affected the program. For example, in the input vector shown in Figure 3, the first element is 1, signifying that the corresponding transformation (loop unroll) was applied and did modify the program. The second element is 0, indicating that Dead Code Elimination

(DCE) has not been applied. The third element is $-1$, suggesting that the inline transformation was applied but had no effect. The output is a vector of the same length as the input, representing the probability that each corresponding transformation will be active. In the example shown, the predictor notes that the inline transformation has been applied without affecting the program, making it unlikely to have an effect if applied again. Consequently, its probability (the third element in the output vector) is close to zero. The predicted dormant probabilities are crucial in the phase-ordering search process, as they guide the algorithm to explore transformations more likely to be activated.

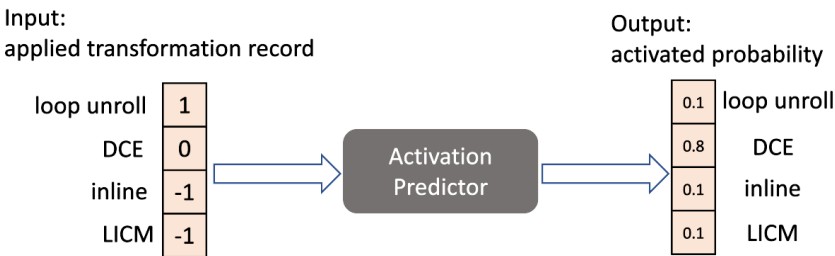

Figure 3: The proposed program representation is used to predict the probability that subsequent transformations will be active.

## 6  FLEXPO

We have implemented a framework, FlexPO, to address the phase-ordering problem. FlexPO integrates the activation predictor into a RL model. We **intentionally** utilize an RL model that is the same or similar to those in other solutions Ashouri et al. (2017); Jain et al. (2022); Mammadli et al. (2020); Cummins et al. (2022), in order to demonstrate the improvements brought about by the activation predictor.

The workflow of FlexPO is illustrated in Figure 4. Initially, the input program is compiled to LLVM Intermediate Representation (IR) **without** applying any transformations, resulting in what we refer to as $raw\_IR$. FlexPO then searches for optimal transformation sequences for $raw\_IR$ across $N$ episodes, each containing $L$ iterations ($N$ and $L$ are hyperparameters). At the beginning of each episode, the current LLVM IR ($curr\_IR$) is set to the initial $raw\_IR$. FlexPO then carries out $L$ iterations sequentially. At the start of each iteration, $curr\_IR$ is provided to the RL agent as an observation. The agent analyzes this observation and selects a transformation $t$. The agent also outputs an estimated value, denoted as $estimated\_reward$, which represents the expected total rewards obtainable by applying the selected transformation. This estimated value is recorded for the purpose of training the agent. Subsequently, FlexPO applies the chosen transformation $t$ to $curr\_IR$, resulting in a new LLVM IR termed $new\_IR$. FlexPO then compiles $new\_IR$ and compares its runtime performance or code size to that of $curr\_IR$. Any improvements are recorded as rewards for training the RL agent. After completing an iteration, $curr\_IR$ is set to $new\_IR$. Once all $L$ iterations are finished, FlexPO updates the RL agent using the recorded data. For RL training, FlexPO employs the generic Proximal Policy Optimization algorithm Schulman et al. (2017).

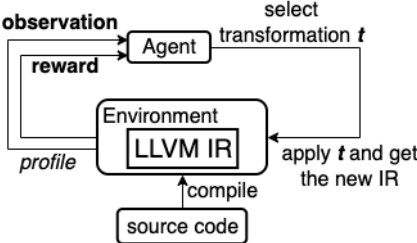

Figure 4: The workflow of FlexPO.

**Observation**: FlexPO employs the method proposed in AutoPhase Haj-Ali et al. (2020) to convert an LLVM IR string into a 56-element vector. Each element is an integrated feature representing a program attribute (e.g., the number of branch instructions, the number of critical edges, the number of

binary operations with a constant operand). All these features are static and can be directly extracted from the IRs without the need for compilation and execution.

**Agent**: FlexPO uses the Actor-Critic approach. In addition to the actor and critic components, the *Agent* also incorporates the activation predictor introduced in Section 5. All three components are implemented as four-layer fully-connected Deep Neural Networks (DNNs) with residual connections He et al. (2016). The structure of the Agent is depicted in Figure 5.

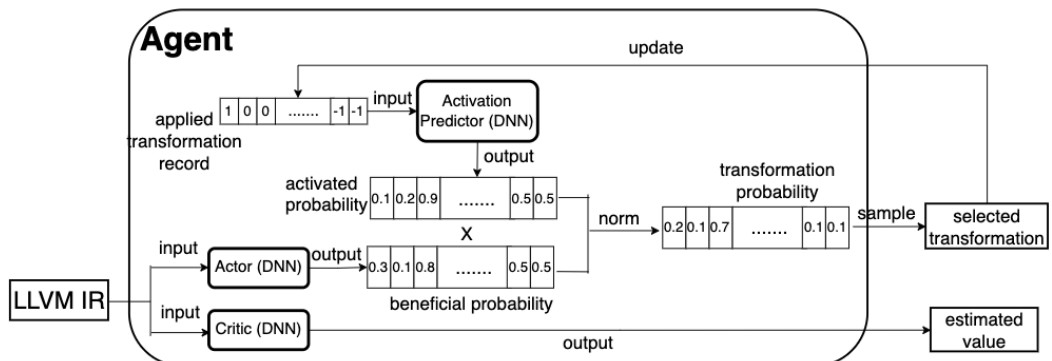

Figure 5: The workflow for the Agent in FlexPO.

- **Critic**: The Critic is responsible for estimating the potential runtime speedup or code size reduction achievable from the current IRs.

- **Actor**: The Actor receives feature vectors generated by AutoPhase and outputs a vector whose length is equal to the number of available actions. This vector represents the probability distribution of all actions. Actions that could yield greater benefits should have higher probabilities. The output from the Actor is multiplied by the output of the Activation Predictor to determine the final probability distribution that is used to sample the subsequent transformations.

- **Activation Predictor**: To enable conservative pruning, the activation predictor is used to predict which transformations are likely to be active. It takes the history of applied transformations as input to predict the dormant status of all subsequent transformations.

**Environment** The environment applies the transformations selected by the Agent and updates the observation. Specifically, it first applies the chosen transformation to the current IR to obtain a new IR. Subsequently, it compiles this new IR to calculate the reward value.

Optimizing for runtime requires the environment to execute the program in each iteration, which incurs a significant overhead. To mitigate this, FlexPO employs caching as an optimization technique. This strategy is based on the observation that during the search process, identical IRs are generated and evaluated multiple times for two primary reasons. First, some transformations are dormant and do not change the IR. Second, certain pairs of transformations are independent of each other; thus, changing their order produces identical IRs. In FlexPO, a database keeps track of the evaluated IRs along with their results. When the environment needs to obtain the runtime for an LLVM IR, it first queries the database to determine if this IR has been previously evaluated. If the IR exists in the database, the stored evaluation result is returned directly. Otherwise, the environment compiles and executes the program, updates the database, and then returns the new results.

**Reward**: For runtime optimization, the reward is the normalized decrease in runtime. The reward of code size is almost the same, except code size instead of runtime is used to calculate the rewards.

## 7 EVALUATION

### 7.1 SETUP

We implemented FlexPO based on CompilerGym Cummins et al. (2022), a toolkit designed for applying Reinforcement Learning to compiler optimization tasks. For benchmarking, we utilized

Ctuning CBench Fursin & Temam (2010), which covers a wide range of applications, including automotive, security, office, and telecommunications. We selected large datasets for each application to minimize variance during measurements. Each program was evaluated five times, and the results were manually analyzed to ensure that both variance and measurement errors were small enough to be negligible. For assessing code size, we measured the size of the code sections in the generated binary files. The programs were evaluated on an 11th Gen Intel Core i7-11700 CPU backend.

## 7.2 ACTIVATION PREDICTOR

In this section, we evaluate the effectiveness of the activation predictor, a crucial component for applying conservative pruning. Since there is no existing dataset suitable for our purpose, we had to generate one ourselves. Specifically, we randomly generated transformation sequences, each with a length of 1000. We then applied these sequences individually and recorded which transformations resulted in changes to the programs. Using this data, we constructed our dataset. Each sample in this dataset consists of a tuple: the first element represents the history of applied transformations; the second element represents the transformation to be predicted; and the third element is the ground truth label indicating whether the transformation is active or not. The process of dataset generation is illustrated in Figure 6.

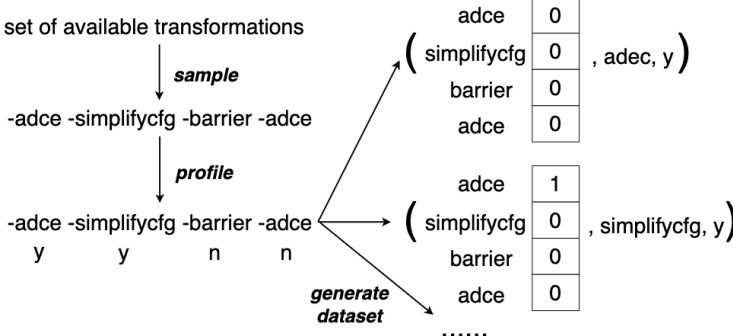

Figure 6: The process of constructing datasets to train the activation predictor.

We generated 921,000 tuples from the qsort program to form the training dataset. For validation, we generated 392,000 and 248,000 tuples from the dijkstra and sha programs, respectively. The activation predictor was trained using the Adam optimizer Kingma & Ba (2014) with a learning rate of $1 \times 10^{-4}$. The learning curve is depicted in Figure 7. The similarity between the training and validation datasets indicates that the activation predictor generalizes well across different programs.

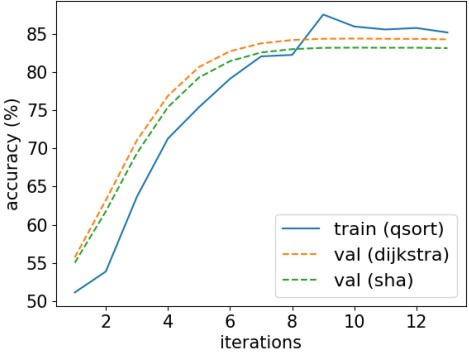

Figure 7: The learning curves for training the activation predictor.

Further, the confusion matrix is provided in Table 1. Our analysis reveals that 78% and 76% of the samples in the dijkstra and sha datasets, respectively, are negative, which implies that most

transformations are dormant. This further substantiates that conservative pruning can effectively reduce the search space.

| Programs | True Positive | True Negative | False Positive | False Negative |
|----------|---------------|---------------|----------------|----------------|
| Dijkstra | 57 693 | 272 344 | 35 115 | 26 848 |
| SHA | 37 516 | 168 456 | 19 813 | 22 215 |

Table 1: The confusion matrix for two validation datasets. The data indicates that most transformations are dormant, highlighting the importance of pruning dormant transformations.

### 7.3 Comparison of FlexPO with -Oz and -O3

Modern compilers offer sets of predefined optimization sequences, among which -O3 and -Oz are the most commonly used. These pipelines are designed to optimize programs for maximum speed and minimal code size, respectively.

In our evaluation, FlexPO primarily seeks sequences to achieve the lowest runtime. To limit the search space, it stops an episode after applying 80 transformations for most programs. However, for the bitcount, patricia, qsort, and tiff2rgba programs, we found that increasing the number of iterations to 350 per episode significantly reduces the runtime. FlexPO conducts the search across 20 episodes, leading to 1600 compilations and executions for most programs. These hyperparameters were empirically determined. The evaluation results are presented in Figure 8. Out of 15 applications, FlexPO outperforms -O3 in 12, producing up to 10x and 7x speedups for stringsearch and bitcount, respectively. On average, excluding the stringsearch and bitcount, FlexPO-compiled programs are 12% faster than those compiled with -O3.

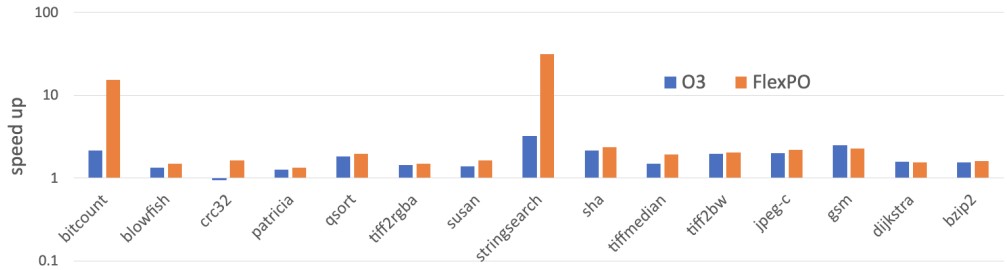

Figure 8: Speedup of programs compiled by FlexPO and -O3, compared with unoptimized versions.

We use stringsearch as a case study to elucidate why FlexPO outperforms -O3. The function, which takes up 99% of the execution time, contains complex control flow. FlexPO discovers a sequence that optimizes the control flow graph, reducing the number of branch instructions. Profiling data, shown in Table 2, reveals that FlexPO-generated code has fewer executed instructions and branch instructions.

| Metric | Unoptimized | -O3 | FlexPO |
|--------|-------------|-----|--------|
| Runtime (sec) | 12.231 | 3.692 | 0.104 |
| # of Instructions | $8.1 \times 10^{10}$ | $3.6 \times 10^{10}$ | $1 \times 10^9$ |
| # of Branches | $1.5 \times 10^{10}$ | $1.0 \times 10^{10}$ | $4.4 \times 10^8$ |
| # of Branch Misses | $5.4 \times 10^7$ | $1.2 \times 10^7$ | $4.5 \times 10^5$ |

Table 2: Profiling data for the stringsearch program.

Similarly, FlexPO was also used to find sequences that minimize code size. On average, FlexPO produces programs with text segments that are 17% smaller compared to those generated by -Oz. Detailed results are provided in the appendix.

### 7.4 Comparison of Aggressive and Conservative Pruning

To demonstrate that **the improvement is brought by the proposed conservative pruning instead of the RL model**, we apply the aggressive pruning and the conservative pruning with the same

RL model. Specifically, we use the sequences proposed in MiCOMP Ashouri et al. (2017) and POSET-RL Jain et al. (2022) for aggressive pruning. MiCOMP clusters the transformations in the O3 pipeline to generate five sequences and searches based on these sequences. Similarly, POSET-RL generates 15 sequences by analyzing the Oz pipeline. In the evaluation, we use the same RL model with different sets of actions: the agent selects **a sequence of transformations** for the aggressive pruning (related works) and selects **a single transformation** for the conservative pruning. We stop an episode when there are 80 applied transformations to make sure the same amount of transformations are applied in aggressive/conservative search. MiCOMP uses five sequences, and their lengths are 36, 2, 3, 1, and 4. Thus, to form a sequence with length 80, there are around $3 * 10^{22}$ possible combinations[1], which is the size of the search space. For POSET-RL, it contains 15 sequences, and the size of the search space is around $9 * 10^{16}$. For FlexPO, the search space is $124^{80}$, which is exponetially larger than MiCOMP and POSET-RL.

In Figure 9(a), we visualize the search process. The figure records the speed up compared with unoptimized programs. The sequences proposed in MiCOMP are generated from the O3 pipeline that contains prior knowledge from human experts. Thus, searching based on these sequences can find good solutions in a few episodes. On the other hand, FlexPO has a large search space and takes more episodes to find optimal solutions. However, given enough amount of search time, FlexPO can always find solutions better than MiCOMP's. The conclusion is the same when comparing FlexPO and POSET-RL for code size improvement (Figure 9(b)).

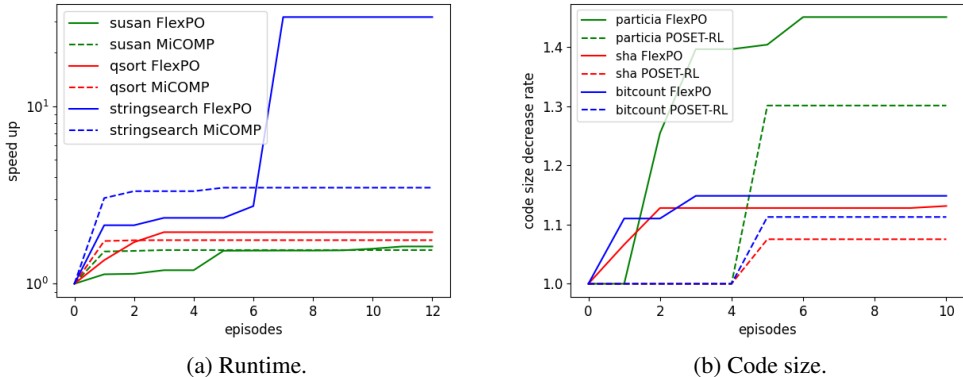

(a) Runtime.                    (b) Code size.

Figure 9: The comparison of conservative pruning (FlexPO) and aggressive pruning solutions (MiCOMP and POSET-RL).

We also evaluate the result of searching with/without the activation predictor, and find that with the activation predictor, FlexPO can find optimal solutions in much fewer episodes/iterations. Additionally, we also evaluate the impact of tuning the hyperparameter (episode/iteration) for the search process. The detailed results are recorded in the appendix.

## 8 CONCLUSION

The phase-ordering problem is challenging due to the large search space. Existing solutions rely on prior knowledge from human experts to aggressively prune the search space, which may exclude optimal solutions and is not scalable to new transformations. In this paper, we propose conservative pruning as an alternative to aggressive pruning. It ensures that the optimal solutions remain within the search space during the pruning process. The insight behind conservative pruning is the prediction of the dormant status of each transformation, focusing the search on transformations that are likely to be activated. Conservative pruning does not rely on human expertise and is scalable to new transformations. We introduce FlexPO, a toolkit that integrates conservative pruning with a RL model to solve the phase-ordering problem. Our experimental results demonstrate that FlexPO generates programs that are 12% faster than those optimized with -O3 and 17.6% smaller than those optimized with -Oz on average.

---

[1]The numbers are calculated by dynamic programming.

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

# A  DETAILED RELATED WORK

The phase-ordering problem is an unsolved problem with a long history. In the early days, some researchers used *predictive heuristics* to directly find an optimal transformation sequence without compiling/executing a program multiple times. However, since the number of transformations increases rapidly Bacon et al. (1994) and the interaction between transformations cannot be precisely predicted Whitfield & Soffa (1997), these methods always remain suboptimal in many cases Triantafyllis et al. (2003). Thus, nowadays most researchers Bodin et al. (1998); Agakov et al. (2006); Ashouri et al. (2016b); Cavazos et al. (2007) use *iterative compilation* to solve the phase-ordering problem. The compilers using iterative compilation optimize programs with different transformation sequences, evaluate the generated code, and select the best one. Compared with predictive heuristics, evaluations are based on the real generated codes, which makes iterative compilation achieve better solutions Triantafyllis et al. (2003).

Machine Learning models are popular choices to implement iterative compilation. Some researchers use Genetic Algorithms (GA) to search the optimal sequences Cooper et al. (1999; 2002); Kulkarni et al. (2004). These methods generate a number of *genes* (transformation sequences) and compile the programs accordingly to evaluate the fitness value for each gene and select the best ones. These methods cannot utilize the information about programs and only regard them as black boxes. Due to the limited information, these algorithms always require a long time to get a good solution even with a small set of transformations. And the search has to be re-executed for new programs.

Some researchers propose solutions to utilize the programs' information during the search. Agakov et al. (2006) form several classical programs into a dataset and use either the Markov model or the Independent Identically Distributed (IID) model to learn the distribution of which passes work well for each program in the dataset. For a new program, the algorithm captures the program's features and finds which program $p$ in the dataset has the closest feature. Then, use the distribution learned by $p$ to sample a sequence as the output. Martins et al. (2016) use a similar method to implement the search process with GA. Their method uses static features including whether the loops have calls and the number of instructions in loops. The static features can be captured directly from the programs, without the need to execute. These features are easily captured but contain limited information. Cavazos et al. (2007) utilize dynamic features (e.g., cache miss, branch miss) to build the model. These dynamic features depend on special hardware, which cannot easily be migrated to other architectures. Some projects Ashouri et al. (2016b;a) use microarchitecture-independent workload characterization (MICA) Hoste & Eeckhout (2007) to make the trained model portable to different hardware. Instead of relying on profiling tools to analyze the programs, some researchers use Deep Neural Networks to directly convert source code into vectors. Cummins et al. (2017) use a Recurrent Neural Network (RNN) to map source code into a fixed-length vector. This method regards code as strings, which do not capture the topology information. Neural Code Comprehension Ben-Nun et al. (2018) proposes a contextual flow graph to capture dataflow information as well. ProGraML Cummins et al. (2020) further captures the control flow information. The outputs of ProGraML are graphs, which need to be trained with Graph Neural Networks (GNN). Instead of handling source code, IR2Vec VenkataKeerthy et al. (2020) maps LLVM IRs to vectors.

The bloom of Machine Learning not only results in more solutions to parse programs but also affects the search process. Kulkarni & Cavazos (2012) train a Neural Network, which accepts the feature vectors for the programs and generates the selected transformations. Ashouri et al. (2016a) use ML models to predict the speed up for a given transformation sequence; Vaswani et al. (2007) use ML models to predict the performance by using the hardware features as inputs. Both methods can avoid the workload to execute programs on real hardware. MiCOMP Ashouri et al. (2017) uses the recommendation system knowledge to explore the search space to avoid the local minima.

As an important area in Machine Learning, Reinforcement Learning also has been used for solving the phase-ordering problem. Mammadli et al. (2020) utilize Deep Reinforcement Learning to solve the phase-ordering problem. Their framework supports searching on different granularities: sequences of transformations (coarse-grained), transformations with default arguments (fine-grained), and transformations and their arguments (finer-grained), but as reported by the authors, the framework can hardly find sequences that significantly surpass O3. POSET-RL Jain et al. (2022) uses RL to search for transformation sequences to reduce the code size. They implement the search process with

the aggressive pruning; they cluster the transformations in the Oz pipeline into 15 or 34 sequences and search on them.

Compared with these related works Ashouri et al. (2017); Jain et al. (2022), FlexPO searches on individual transformations instead of sequences. Thus, FlexPO has a larger search space. Although the framework in Mammadli et al. (2020) also searches on individual transformations, the framework finds sequences that surpass the LLVM O3 pipeline only around 3% in a reasonable time. This is due to it doesn't apply pruning during searches. Instead, FlexPO applies conservative pruning and finds sequences 12% better than the LLVM O3 pipeline.

# B    EVALUATION RESULT

## B.1    FLEXPO VS -OZ

The same as the runtime, we use FlexPO to search for sequences that get the smallest code size (the number of bytes in the text segment). We set $episode$ to 15 and $iteration$ to 150. Other hyperparameters are the same as the runtime experiments. The result is shown in Figure 10. On average, FlexPO can generate programs that have 17% smaller text segments compared with Oz.

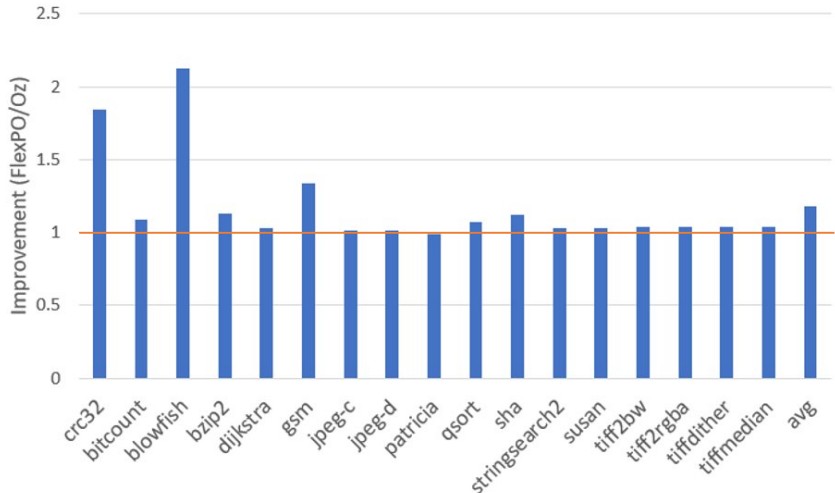

Figure 10: The code size improvement. The value is computed as $\frac{Oz\ code\ size}{FlexPO\ code\ size}$.

## B.2    VALIDATION OF THE ACTIVATION PREDICTOR

In this section, FlexPO is used to search optimal sequences with or without the pretrained activation predictor. The activation predictor is used to avoid exploring transformations that are dormant. Thus, it is helpful for getting the optimal solutions with fewer searches.

To evaluate the ability of the activation predictor to avoid exploring dormant transformations, we execute FlexPO 10 episodes, with each episode having only 10 iterations. We visualize the search process in Figure 11 for three applications (susan, qsort, stringsearch). Without the activation predictor, the highest improvements are close to 1.0, which means the transformed LLVM IRs have close performance with the unoptimized LLVM IRs. This indicates that the transformations explored by FlexPO w/o the predictor do not change the LLVM IR significantly. Instead, with the activation predictor, FlexPO avoids exploring these dormant transformations. Thus, FlexPO can significantly improve the runtime by applying only 10 transformations.

## B.3    COST FOR SEARCH

Finally, we analyze the relationship between search time and the search result. The search time depends on the following four factors: the number of episodes $E$, the number of iterations $L$, the

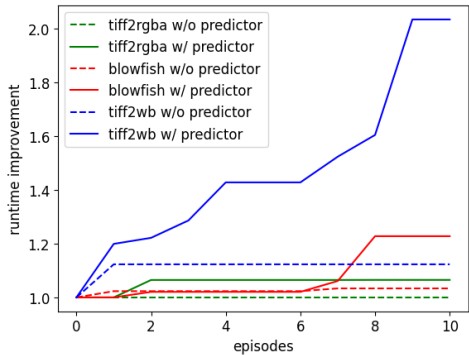

Figure 11: The search process with/without the activation predictor.

time to get reward information $R$, and the training process $T$. The amount of time can be roughly calculated as $E * L * R + E * T$. Since FlexPO uses lightweight DNNs, $T$ is much smaller than $R$. Thus, the amount of time can be regarded as $E * L * R$.

For a given program and fixed inputs, the $R$ is a constant. For code size improvement, $R$ is the time to compile the program, while for runtime improvement, $R$ includes the execute time additionally.

Both $E$ and $L$ are hyperparameters. $E$ is the number of times to search from the unoptimized IRs (explore), while $L$ is the depth for each search process (exploit). For FlexPO, each iteration selects a transformation. Thus, $L$ is also the maximum length of the transformation sequence.

First, we set episode $E$ to 10 and search with different $L$. Then, we set iterations $L$ to 60 and search with various $E$. The experiment results are shown in Figure 12(a) and Figure 12(b) respectively. As we can conclude, generally, FlexPO finds better solutions with larger iterations or episodes. However, the number of compilations and executions ($E * L$) increases linearly with the episode or iteration. Thus, the hyperparameter configuration is a knob for users to decide the trade-off between search time and output quality.

In our evaluation, we find $E = 20$ and $L = 80$ are sufficient to find solutions that outperform O3/Oz for most applications. The evaluation in Section 7.3 uses this configuration and can find sequences that surpass O3/Oz. The dijkstra application has the longest execution time (the largest $R$), thus, it also has the longest search time. With $E = 20$ and $L = 80$, the search takes around 66 minutes. However, as discussed in Section 6, some IRs are repeatedly shown up during the search, and FlexPO only executes them the first time it shows up. Thus, in real situations, the search time is much shorter, around 15 minutes in our evaluation.

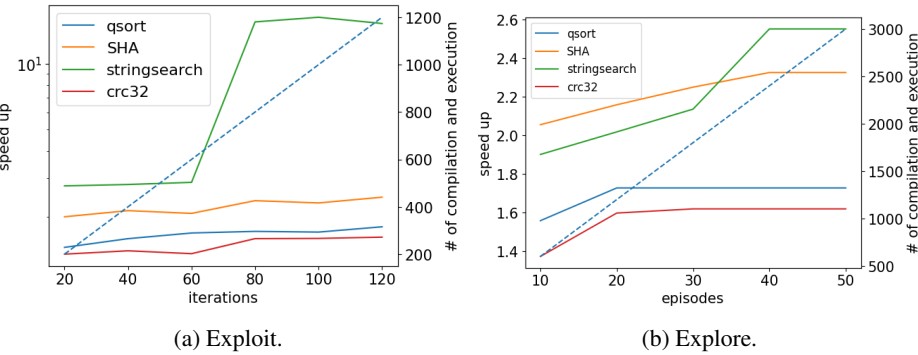

(a) Exploit.    (b) Explore.

Figure 12: In general, FlexPO finds better solutions with larger episodes (explore) or iterations (exploit). However, the cost of the search increases linearly with episodes or iterations.

