# OpenReview forum: "Exponentially Expanding the Compiler Phase-Ordering Problem's Search Space through the Learning of Dormant Information"
_ICLR.cc/2024/Conference — ICLR 2024 Conference Withdrawn Submission_

### Official Review · Reviewer_5iQh · 2023-11-01

**Soundness:** 2 fair
**Presentation:** 2 fair
**Contribution:** 1 poor
**Rating:** 3
**Confidence:** 3

**Summary:**

The authors introduce FlexPO, a reinforcement learning transformation selector for program compilation. FlexPO works by selecting a sequence of program transformations that improves the observed runtime of a program. Unlike prior work, FlexPO employs an activation predictor, which estimates which transformations might be fruitfully applied in the future.

**Strengths:**

FlexPO tackles an important problem of improving program compilation. A massive amount of effort goes into designing existing compiler pipelines, and we already know that they fall short of optimal. Using ML techniques to help is a great idea.

**Weaknesses:**

Overall, two large points confused me about this submission:

1) The authors seem to ignore the cost of the actual compilation procedure. This is fine, and has been studied under the name "super optimization" for a long time. But it seems to me that comparing FlexPo, which is a profile-guided, multi-shot, expensive optimizer, with standard O3, which is a heuristic-guided, single-shot, and "fast" optimizer, doesn't make much sense. At a minimum, the authors should report and compare the compilation time of their approach. At best, the authors should compare with other approaches that use a similar amount of time.

2) I do not understand the motivation of predicting which transformations are dormant or active. It seems this is not ablated in the experiments either -- perhaps the authors could clarify why this prediction is difficult, and include an experiment using ground-truth dormant/active information?

**Questions:**

Q1) I don't understand what F1 is showing me. Shouldn't the x-axis be time? I imagine a very naive solution could consider a very large search space, it just couldn't do it effectively.

Q2) "However, their framework faced difficulties... to the expansive search space" -- what are the difficulties? what are the times?

Q3) After S3, I still do not know if the authors are seeking a single, global order that works for all programs or are generating an "instance-optimized" order-per-program.

Q4) Can you explain why it is difficult to determine whether or not a transformation will induce a change to the underlying IR? It seems a very simple strategy would do the trick: "clone the current IR, apply the transformation in question, and then see if anything changed." Sure, it's O(n) in the size of the program, but so is constructing the feature representation described in S6...

Q4) The text right above F2 makes it seem like each entry in the feature vector is going to have three states: whether the transformation has been applied, and, if the transformation has been applied, whether or not the transformation is now dormant (you  call these 1, 0, and -1 in S4). But F2 only shows 2 states.

Q5) I'm confused about what the activation predictor does up until S6. First, the authors say that their ML model is going to predict whether or not a transformation is active or dormant. Then, the authors say that the features used as inputs to the model will be... whether or not a transformation is active or dormant. At first glance, this seems like quite the easy problem! :D After reading S6, I (think I) understand that the activation predictor estimates which transformations will *still* be active after applying all *currently* active transformations.

Q6) S6 introduces additional features used by the RL agent that are not those described in S5... this is the first time it becomes clear to me that the activation predictor is going to be a component inside of an existing RL system. After reading S6, I still cannot say for certain if the activation predictor is trained inside the RL loop or ahead of time in a supervised fashion.

Q7) In S7.3, you compare FlexPO with a certain number of itreations to the existing optimization modes. But FlexPO has a huge advantage in that the agent gets to observe the runtime delta as a reward function. O3 and Oz, as far as I know, use cost models, not actual execution. At best, this is a comparison between a traditional optimizer and a profile-guided optimizer. At worst, I imagine that FlexPO takes *significantly* longer to run than O3 and Oz! Heck, even executing a search routine *one time* on an input large enough to be interesting is probably 10x more expensive than the compilation procedure... Perhaps you could consider FlexPO a type of "superoptimizer?"

Q8) At a minimum, you should plot the results are Pareto scatter plots, with compilation time on one axis and runtime on the other. If you looked into how existing GCC/Clang O3 works, and gave it the same amount of time to try transformations as FlexPO, what would the results be?

Typo: "Mammadli et al. Mammadli et al. (2020)" and "Kulkarni et al. Kulkarni et al. (2004) "

---

### Official Review · Reviewer_gVMu · 2023-11-01

**Soundness:** 1 poor
**Presentation:** 2 fair
**Contribution:** 2 fair
**Rating:** 3
**Confidence:** 4

**Summary:**

FlexPO uses reinforcement learning and the status of optimization steps to address the phase ordering problem. Unlike common practices, FlexPO doesn't limit optimization options based on human knowledge. Instead, it tracks which optimizations are applied and whether they are active or dormant, and uses them to represent program status. This approach leads to better speedup and smaller program sizes compared to -O3 and -Oz flags.

**Strengths:**

- It shows that RL w/ the activation predictor to predict dormant status can be effective in tackling the phase-ordering problem

**Weaknesses:**

- It claimed to introduce a novel program representation based on the dormant status of applied optimizations, but it seems the representation is not used in the pipeline as observation. Also, it is unclear how effective the proposed dormant representation is compared to the other program representations.
- Its evaluation needs to be improved. RL is not an established SoTA for tackling the problem. It would be critical to compare FlexPO to random, Genetic Algorithm, and other algorithms. It would be helpful to show curves from a more comprehensive set of programs (averaged) in Fig. 7,9,11. The current results appear to be selectively chosen..
- Not relying on human knowledge is considered a fundamental feature in data-driven machine learning. However, it's worth noting that harnessing human knowledge can be an efficient approach to reducing search complexity. I believe that certain dormant relationships can be statically inferred from the optimization implementation or learned just once.

**Questions:**

1. Can you please clarify how you determine an optimization is dormant? Do you compare the IR of the program before and after the optimization?
2. Can you elaborate on how you integrate the dormant representation with your observation? In observation, you only mentioned the vector features that directly extracted from the programs.

---

### Official Review · Reviewer_voGU · 2023-11-04

**Soundness:** 3 good
**Presentation:** 2 fair
**Contribution:** 2 fair
**Rating:** 5
**Confidence:** 4

**Summary:**

The paper aims to learn the "dormant" transformations such that it does not have to rely on human expertise. The word dormant is used throughout the paper to describe cases where multiple calls of a certain optimization pass does not incur any more improvement.

The paper first dives into a program representation: a vector of active/dormant status of applied transformations. The paper assumes that the combination of the active/dormant status in the vector helps determine various characteristics of the program. Diving into more details about the vector, it seems that there are 3 elements (1: transform was applied + had effect, 0: transform was not applied, -1: transform was applied + had no effect).

The paper describes the input and output of the policy network where the input is the above representation from a run, and output is the likelihood of each transformation pass having impact in the subsequent run. The paper naturally drops this into a reinforcement learning environment.

The paper experimented the accuracy of activation predictor, end-to-end comparison between FlexPO and Oz/O3 (claims 12% faster than O3). The paper attributes this to fewer executed instructions and branch instructions.

Overall, I do like the paper, but there seems to be some weaknesses to be addressed.
I would like to re-evaluate after rebuttal.

**Strengths:**

+ The paper presents a very simple yet neat way of representing a program: vector of transformations.

**Weaknesses:**

- It is difficult to understand some of the figures.
- The paper lacks the design details of the FlexPO besides its high level design. Seems to make it very difficult to reproduce.
- Comprehensive set of results for runtime/code size against other approaches seem to be missing. IMHO it is very difficult to judge if the FlexPO has superior performance versus other approaches. It would be of more help if the authors can present in which benchmarks it performed better & why.

**Questions:**

* Please provide relation between the elements of Fig. 4 and Fig. 5, as it would be helpful to better understand the design.
* Was the 56-element vector following AutoPhase include all the information? For example, wouldn't the graph approach in Cummins et al. 2020 complementary to the work?
* Can you provide the compilation time overhead of the approach for big programs? In some real scenarios, we want very fast compilation without compromising on the runtime performance.
* Can you provide more details about the FlexPO design? Size of the network, how some of the design decisions were made etc. It would be a great addition for the future researchers that build on this work.

---

### Official Review · Reviewer_wMoc · 2023-11-11

**Soundness:** 3 good
**Presentation:** 4 excellent
**Contribution:** 4 excellent
**Rating:** 8
**Confidence:** 3

**Summary:**

The phase-ordering problem in compiler optimization, which is to find an optimal sequence of compilation transformations, is constrained by the vast number of potential transformation combinations. Traditional methods address this by *aggressively* pruning the search space using expert knowledge, but this risks excluding optimal solutions and lacks scalability. This paper introduces a more *conservative* approach using machine learning, which only eliminates non-optimal solutions, thus remaining adaptable to new transformations. The proposed solution, FlexPO, integrated with a reinforcement learning model, can explore an exponentially larger search space, resulting in programs that are up to 12% faster or 17.6% smaller than those generated by contemporary compilers.

**Strengths:**

* Applying "learning" domain information is a novel and effective strategy.
* The resulting compilation framework could be highly beneficial for both researchers and practitioners.

**Weaknesses:**

* The paper lacks a comparative evaluation of the quality of dormant transformation prediction with existing works, such as [Kulkarni et al. 2004].

**Questions:**

Thank you for submitting to ICLR 2024. The paper presents an interesting approach to reducing the search space by leveraging dormant information. While this is not the first work to utilize such information in solving the phase-ordering problem, its application to reduce the search space seems novel and effective, potentially yielding higher-quality code than -O3 and -Oz. My questions are as follows:

* What is the typical range of compilation time when using FlexPO?
* Considering that this work isn't the first to utilize dormant information in compiler transformation, a comparison with existing predictors like [Kulkarni et al. 2004] in terms of prediction quality would have been beneficial. How does FlexPO’s quality of prediction compare?
* In Section 5, the effect of a transformation (1) or its lack (-1) generally depends on which transformations are applied beforehand, which is essentially the phase-ordering problem. However, in Figure 3, the input to the activation predictor is a simple bit vector that doesn't capture this aspect, and the model appears to predict the dormant probability without considering this ordering information. How does FlexPO account for this?
* In Section 6, FlexPO utilizes a database to reduce evaluation time for duplicate IRs. Could you elaborate on how this cache is implemented? Does it involve bitcode-by-bitcode comparison, or does it employ a hashing mechanism or something else to expedite comparison?